# Identification and experimental validation of *BMX* as a crucial PANoptosis-related gene for immune response in Spinal Cord Injury

Tianbao Feng[1,2], Jiating Hu[2], Mi Xie[2], Guodong Shi[2], Qi Wang[2], Jingyuan Yao[2], Xiaoqin Liu[2]*

**1** Department of Radiology, The Affiliated Hospital of Yan'an University, Yan'an, Shaanxi, China, **2** Yan'an Medical College of Yan'an University, Yan'an, Shaanxi, China

* liuxq@yau.edu.cn

## Abstract

Spinal cord injury (SCI) is a debilitating neurological condition that severely impacts motor, sensory, and autonomic functions, leading to significant challenges in patient quality of life and imposing substantial economic burdens on society. PANoptosis is an emerging concept in programmed cell death that combines three key processes: pyroptosis, apoptosis, and necroptosis. Research has demonstrated the significant roles of apoptosis, necroptosis, and pyroptosis in the progression of SCI. As such, targeting PANoptosis-related genes may offer new therapeutic targets and clinically relevant treatment strategies. This study seeks to identify distinct molecular subtypes of SCI and potential drugs for its treatment, based on the mechanisms of PANoptosis. We acquired RNA sequencing data from the Gene Expression Omnibus (GEO) datasets GSE151371 and performed Gene Set Variation Analysis (GSVA) and Gene Set Enrichment Analysis (GSEA) analysis to delineate differential biological functions between SCI patients and healthy controls. We identified a total of 1138 significant differentially expressed genes (DEGs), comprising 431 downregulated and 707 upregulated genes. We intersected DEGs with PANoptosis gene sets and identified 23 common genes. 23 PANoptosis-related genes were subjected to functional enrichment analysis and PANoptosis scores calculation. PANoptosis score in SCI samples was significantly higher than in HC samples. Additionally, a protein-protein interaction (PPI) network was established to identify hub genes, and 8 machine learning algorithms were used to narrowed down hub genes. *BMX* and *CASP5* were consistently identified across all algorithms. Immune cell infiltration analysis revealed significant correlations between *BMX* and several immune cell types, highlighting its involvement in the inflammatory response after SCI. Through additional ROC curve analysis, we confirmed the promising diagnostic potential of *BMX*, with an AUC value of 0.987. Moreover, we predicted potential therapeutic agents and key regulatory factors interacting with *BMX*. We performed single-gene GSEA analysis to explore the biological

**Data availability statement:** The dataset analyzed during the current study are available in the Gene Expression Omnibus repository (https://www.ncbi.nlm.nih.gov/geo/) (accession number: GSE151371).

**Funding:** The author(s) received no specific funding for this work.

**Competing interests:** The authors have declared that no competing interests exist.

functions and pathways associated with *BMX*. Finally, we created a rat model of SCI to experimentally confirm the elevated expression of BMX in the SCI group by quantitative real-time PCR (qRT-PCR), western blot (WB) and immunohistochemistry (IHC). In conclusion, our findings provide valuable insights into the molecular mechanisms underlying SCI, highlighting *BMX*, a PANoptosis-related gene, as a potential therapeutic target. These results underscore the necessity for future studies to explore these targets in clinical applications.

## Introduction

Spinal cord injury (SCI) represents a significant health challenge, characterized by complex pathophysiological processes leading to profound neurological deficits and a substantial impact on patients' quality of life [1,2]. The pathophysiological processes of SCI are highly complex, involving a cascade of events such as inflammation, apoptosis, and neurodegeneration, which complicate the development of effective treatments [3,4]. The economic burden of SCI extends beyond the patient, with substantial costs to healthcare systems and society [5]. Current treatment options, including surgical intervention, pharmacological treatments, and rehabilitation, often fail to provide significant functional recovery, particularly for motor and sensory functions [6]. This highlights the urgent need for continued exploration into the molecular pathways involved in SCI and for the identification of new potential therapeutic targets.

A promising avenue of research involves PANoptosis, a complex form of programmed cell death that integrates apoptosis, pyroptosis, and necroptosis [7]. PANoptosis has been implicated in a variety of diseases, including infectious diseases, cancers, neurodegenerative disorders, and inflammatory diseases [8,9]. Although its role in SCI is not yet fully understood, emerging evidence suggests that PANoptosis may be crucial in modulating neuroinflammation and cellular injury, both of which play pivotal roles in the pathogenesis of SCI [10].

In this context, this study aims to examine the involvement of PANoptosis-related genes in SCI. By analyzing gene expression data from SCI patients and healthy controls, we seek to identify key genes involved in PANoptosis and explore their functional roles in SCI pathology. Understanding the molecular mechanisms underlying PANoptosis in SCI could reveal new biomarkers and therapeutic targets, paving the way for more effective interventions and better clinical outcomes for SCI patients.

## Materials and methods

### Data acquisition and processing

In this study, RNA transcriptome data were obtained from the public Gene Expression Omnibus (GEO, https://www.ncbi.nlm.nih.gov/geo/). The chosen gene microarray dataset was GSE151371, with Homo sapiens as the designated species and peripheral white blood cells as the specified tissue type [11]. The dataset comprised information from 10 healthy control patients free from spinal cord injuries (HC group) and 38 SCI patients (SCI group). To complement our spinal cord tissue analysis in

rats, we selected the GSE151371 dataset, which includes RNA-seq profiles from peripheral blood leukocytes of human SCI patients and HC controls. Although the tissue types differ, this dataset provides valuable insights into the systemic transcriptomic response to spinal cord injury. Gene expression data were normalized using the "normalizeBetweenArrays" function in the "limma" package of R software.

## GSVA enrichment analysis

GSVA and functional enrichment analysis were performed using the "GSVA" package to explore differences between the disease group and the control group [12]. The "h.all.v2024.1.Hs.symbols.gmt" file was downloaded from the MsigDB for subsequent pathway enrichment analysis [13]. The GSVA scores of the two groups were compared using the "limma" package to determine differential biological functions and expression pathways [14].

## Screening of differentially expressed PANoptosis-related genes

A total of 508 unique PANoptosis-related genes, including those involved in apoptosis, pyroptosis, and necroptosis, were obtained from the literature after removing duplicates and are listed in S1 Table for subsequent analysis [15–17]. Differential analysis of the dataset was conducted using the "limma" package with a filter condition of |log2FC|>0.5 and adj.$P<0.05$. The intersection of the differentially expressed genes and 508 PANoptosis-related genes yielded 23 PANoptosis-related differentially expressed genes (DEGs). These 23 genes were visualized using the "heatmap" package and mapped on chromosomes using the "RCircos" package.

## PANoptosis score calculation

To identify PANoptosis-related genes, DEG analysis was conducted for different PANoptosis subtypes using the limmaR package, with important features selected based on adj.$P<0.05$ and a |log2 fold-change (FC)|>0.5. Based on PANoptosis-related gene features, patients were clustered into different groups for further analysis. Principal Component Analysis (PCA) was then performed using principal components 1 and 2 to determine PANscore [18]. The PANscore calculation formula was: PANscore=∑ (PC1i+PC2i), where i is the expression of differentially expressed PANoptosis-related genes.

## Protein-Protein Interaction (PPI) network analysis

PPI interaction analysis was performed using the STRING protein database (https://cn.stringdb.org/) [19]. Multiple gene symbols for differentially expressed genes (DEGs) were input into the database. PPI interaction files with a low confidence score ≥ 0.15 were downloaded. Cytoscape software (version 3.9.1) was used to construct and visualize the network. The top 15 significant genes were selected through five algorithms from the cytoHubba plugin (BottleNeck, Closeness, Degree, Betweenness, Stress) [20].

## Functional enrichment analysis

Gene Ontology (GO) and Kyoto Encyclopedia of Genes and Genomes (KEGG) pathway enrichment analyses were conducted on the DEGs using the clusterProfiler package [21].

## Machine learning

To determine the most suitable biomarkers and hub genes for diagnosing SCI, we applied 8 machine learning algorithms: least absolute shrinkage and selection operator (LASSO), support vector machine-recursive feature elimination (SVM), Bagged Trees, Bayesian, Wrapper (Boruta), Learning Vector Quantification (LQV), xgboost, and random forest for further screening. The interactions among these key intersecting genes were then visualized using the "Upset" application in R. The genes identified by all 8 machine learning algorithms were selected for further analysis.

### ROC curve of the hub gene

The ROC curve was employed to further assess the hub gene, with the goal of evaluating the diagnostic potential of the *BMX* gene. The area under the curve (AUC), sensitivity, and 1-specificity were calculated using the "pROC" R package. AUC values of 0.5–0.7/0.7–0.9/>0.9 were recognized as low/medium/high accuracy, respectively. Sensitivity and 1-specificity were used together to evaluate model authenticity, with higher values indicating better model authenticity.

### Pathway analysis of the hub gene

To explore the regulatory pathways and biological functions associated with the hub gene, single-gene GSEA analysis was performed using the "GSVA" package. Adjusted $P<0.05$ was considered significant. The top five enriched pathways in upregulated and downregulated processes were displayed.

### Evaluation of tissue-infiltrating immune cells

Based on expression data, the relative abundance of immune infiltrating cells in SCI and normal samples was estimated using the CIBERSORT package [22]. The proportion of immune cells and their differences between the HC and SCI groups were displayed using box plots. The correlation heatmap of 22 immune cell infiltrations was visualized using the "corrplot" R package. Finally, the correlation between core genes and infiltrating immune cells was visualized using the "ggplot2" package. $P<0.05$ was considered significant. The relationship between gene expression and immune cell infiltration was evaluated by Spearman analysis, and the correlation between BMX gene and activated mast cells was further assessed.

### Prediction of potential therapeutic drugs, transcription factors, and RBP proteins for the hub gene

Potential therapeutic drugs targeting the two feature genes were predicted through the CTD database (https://ctdbase.org/) [23]. RNA-binding proteins (RBP) interacting with the two feature genes were predicted using the StarBase database (https://rnasysu.com/encori/) [24], and transcription factors (TFs) interacting with the feature genes were predicted using the ChIPBase database (https://rnasysu.com/chipbase3/index.php) [25]. The results were visualized using Cytoscape software (version 3.9.1).

### SCI rat model establishment

Female Sprague–Dawley (SD) rats (220–240 g) aged 8 weeks were purchased from the Laboratory Animal Center of Xi'an Jiaotong University (Animal license number: SCXC (Shan) 2023–002). All animal experiments were approved by the Animal Ethics Committee of Yan'an University. To create a severe spinal cord contusion injury at the thoracic level (T10), a standardized procedure was utilized, in accordance with previously established methods [26]. Briefly, rats were anesthetized with 1% pentobarbital sodium (40 mg/kg, intraperitoneal injection). A midline incision was performed on the rat's back at the T9-11 vertebral level to expose the T10 vertebral plate. The SCI group was subjected to treatment using the HI-0400 (PSI) spinal cord percussion device. The rats were positioned in a prone stance on the device's tabletop, with the impact force adjusted to 200 kdyn/cm$^2$. In the Sham group, only the laminectomy was performed without spinal cord injury. SCI was successfully induced in rat, and spinal cord samples were harvested 3 days post-surgery for subsequent analysis. At the end of the experiments, the animals were euthanized by administering an overdose of isoflurane for approximately 10 minutes, followed by exsanguination. The research adhered to the ARRIVE guidelines, and all the procedures outlined in this section were conducted in accordance with the pertinent guidelines and regulations.

### Total RNA extraction and qRT-PCR

After heart perfusion with precooled saline, spinal cord tissue was collected in an ice box. Total RNA was extracted from spinal cord tissue using TRIzol (15596026CN, Invitrogen, USA) according to the manufacturer's protocol. 1 mL of TRIzol

was used for RNA extraction per 20 mg of spinal cord tissue. Total RNA was reverse transcribed into complementary DNA using a reverse transcription kit (K1691, Thermo Fisher Scientific, USA). Real-time quantitative PCR was performed using SYBR qPCR Master Mix (R323-01, Vazyme, China). β-actin was used as an internal reference gene, and relative gene expression levels were quantified using the $2^{-\Delta\Delta Ct}$ method. Primer sequences are shown in table (Table 1).

### Immunohistochemistry (IHC)

Spinal cord sections were harvested and subjected to immunofluorescence staining. The spinal cord tissue samples were initially fixed in paraffin and subsequently sectioned into 5-μm thick slices. The tissue samples were initially treated with 3% $H_2O_2$ for 15 minutes, followed by incubation in a blocking solution for 1 hour. Next, the samples were incubated overnight at 4°C with a primary antibody (rabbit anti-BMX, 1:200, bs-2765R, Bioss, USA). Afterward, the samples were exposed to a secondary antibody conjugated to HRP for 1 hour at 37°C. The reaction was developed using 3,3-diaminobenzidine (DAB, ZLI-9017, ZSGB-Bio, China), and the samples were stained with hematoxylin. Finally, the stained samples were visualized using a laser scanning confocal microscope (Nikon).

### Western blot (WB)

The fresh spinal cord tissue samples were obtained, and total protein was isolated for Western blotting. The membranes were first blocked with 5% nonfat milk in tris-buffered saline with tween 20 (TBST) at room temperature for 1 hour. After blocking, the membranes were incubated overnight at 4°C with primary antibodies targeting Anti-BMX (rabbit anti-BMX, 1:500, bs-2765R, Bioss, USA) and GAPDH (rabbit anti-GAPDH, 1:3000, AF7021, Affinity, China). The next day, the membranes were exposed to HRP-conjugated secondary antibodies (1:5000, A21020, Abbkine, China) for 1 hour at room temperature. Protein bands were then detected using the BeyoECL Plus system (Beyotime, P0018), and the band intensities were quantified using ImageJ software.

### Statistical analysis

All analyses were performed using R version 4.2.1 (https://www.Rproject.org). Data are expressed as means ± standard error of the mean (SEM) from at least three independent experiments and were analyzed with GraphPad Prism 9.3.1. The Wilcoxon rank sum test and Welch's t-test were applied to compare expression differences between unpaired samples of the two groups. Significance levels were defined as *, $P < 0.05$, **, $P < 0.01$, ***, $P < 0.001$, and ****, $P < 0.0001$ for all analyses.

## Results

### Differential gene expression and pathway enrichment analysis in SCI and HC groups

The study flow is shown in Fig 1. The dataset GSE151371 was normalized, and the results before and after normalization are shown in Fig 2A. Under the criteria of adjusted $P < 0.05$ and |log2FC| > 0.5, 1916 DEGs were found between 38 SCI samples and 10 healthy control samples, including 707 upregulated genes and 431 downregulated genes. The volcano plot of typical DEGs visually showed the results of differential expression analysis (Fig 2B). GSEA analysis was performed, and

**Table 1. Primer sequences of *Bmx* and β-actin.**

| Gene | Sequence (5′–3′) |
| --- | --- |
| *Bmx* | Forward: GGCAGAAGAAGCCTAAATGGAC |
| | Reverse: CCTTTTCTGCTTCCTCGTTTCATT |
| β-actin | Forward: CCCATCTATGAGGGGTTACGC |
| | Reverse: TTTAATGTCACGCACGATTTC |

the results showed significant enrichment of pathways related to immune regulation, including cytokine-cytokine receptor interactions, immature neutrophil signaling pathways, primary immunodeficiency, neutrophil degranulation, and B lymphocyte progenitor cell signaling pathways (Fig 2C). Subsequently, GSVA analysis was conducted to study functional changes between the two groups. We found that cholesterol homeostasis, reactive oxygen species signaling pathways, IL-6/JAK/STAT3 signaling pathways, glycolysis, and inflammation-related signaling pathways were upregulated in SCI, while ultraviolet response pathways and pancreatic β-cell pathways were upregulated in the HC group (Figs 2D and 2E).

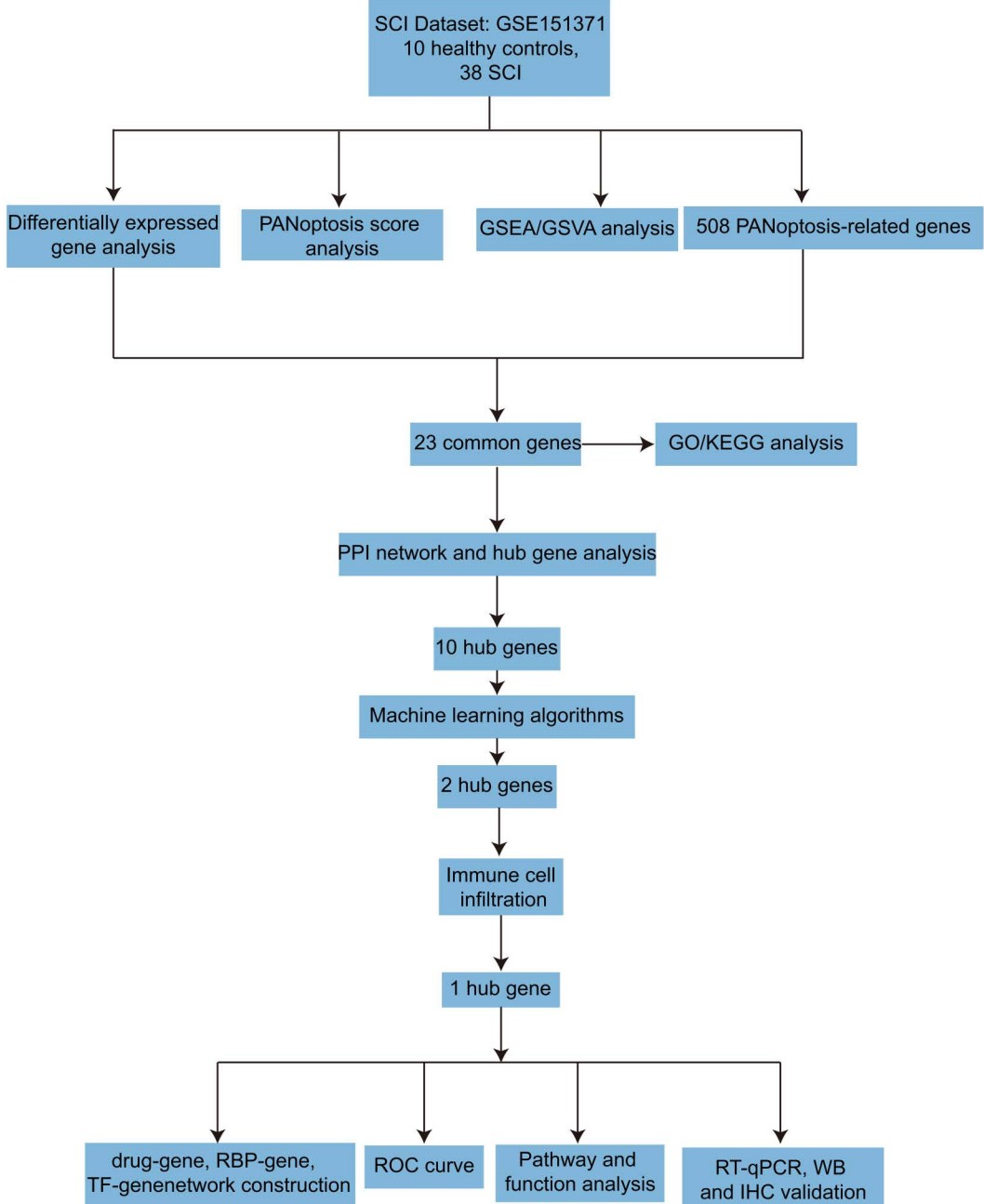

**Fig 1. The overall framework of this study.**

                                           

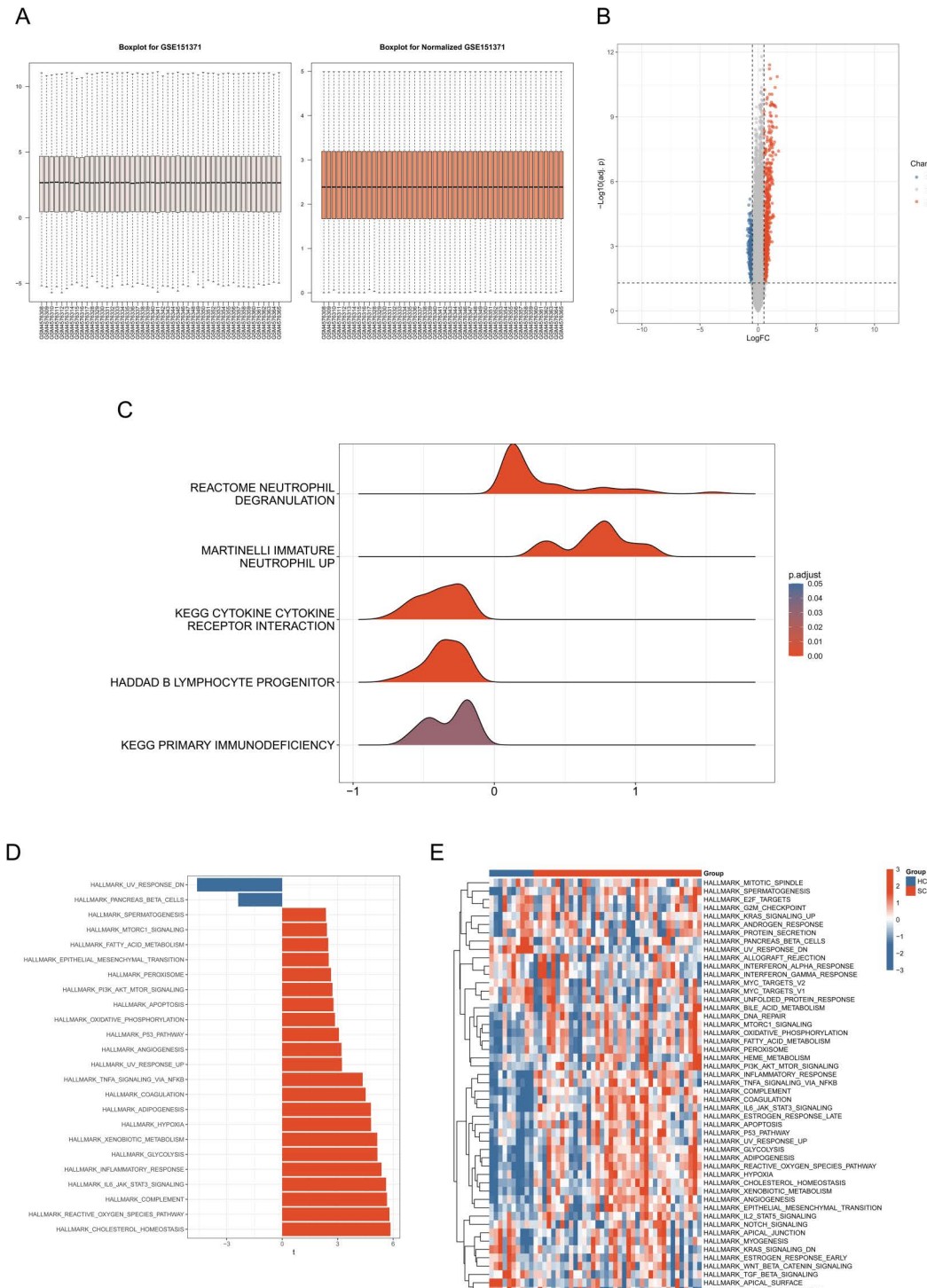

**Fig 2. Identification of different genes and pathways between SCI and HC group. (A)** Boxplots of the GEO dataset distribution before and after normalization. **(B)** Volcano plot of DEGs, where red indicates upregulated DEGs, grey represents genes with no significant difference, and blue indicates downregulated DEGs. **(C)** Ridgeline plots display immune-related pathways from GSEA. **(D)** GSVA analysis of upregulated and downregulated pathways. **(E)** Heatmap showing the pathways analyzed by GSVA in SCI and HC samples.

## Identification of PANoptosis-related DEGs in SCI and their expression and PANoptosis score analysis

We intersected DEGs with PANoptosis gene sets and identified 23 common genes, which were defined as PANoptosis-related DEGs associated with SCI (Fig 3A). Using the 'RCircos' package, the chromosomal locations of these 23 PANoptosis-related genes were mapped (Fig 3B). The expression levels of these 23 genes in SCI and control groups are shown in (Fig 3C). Based on the expression of these 23 PANoptosis-related genes, we calculated the PANoptosis score

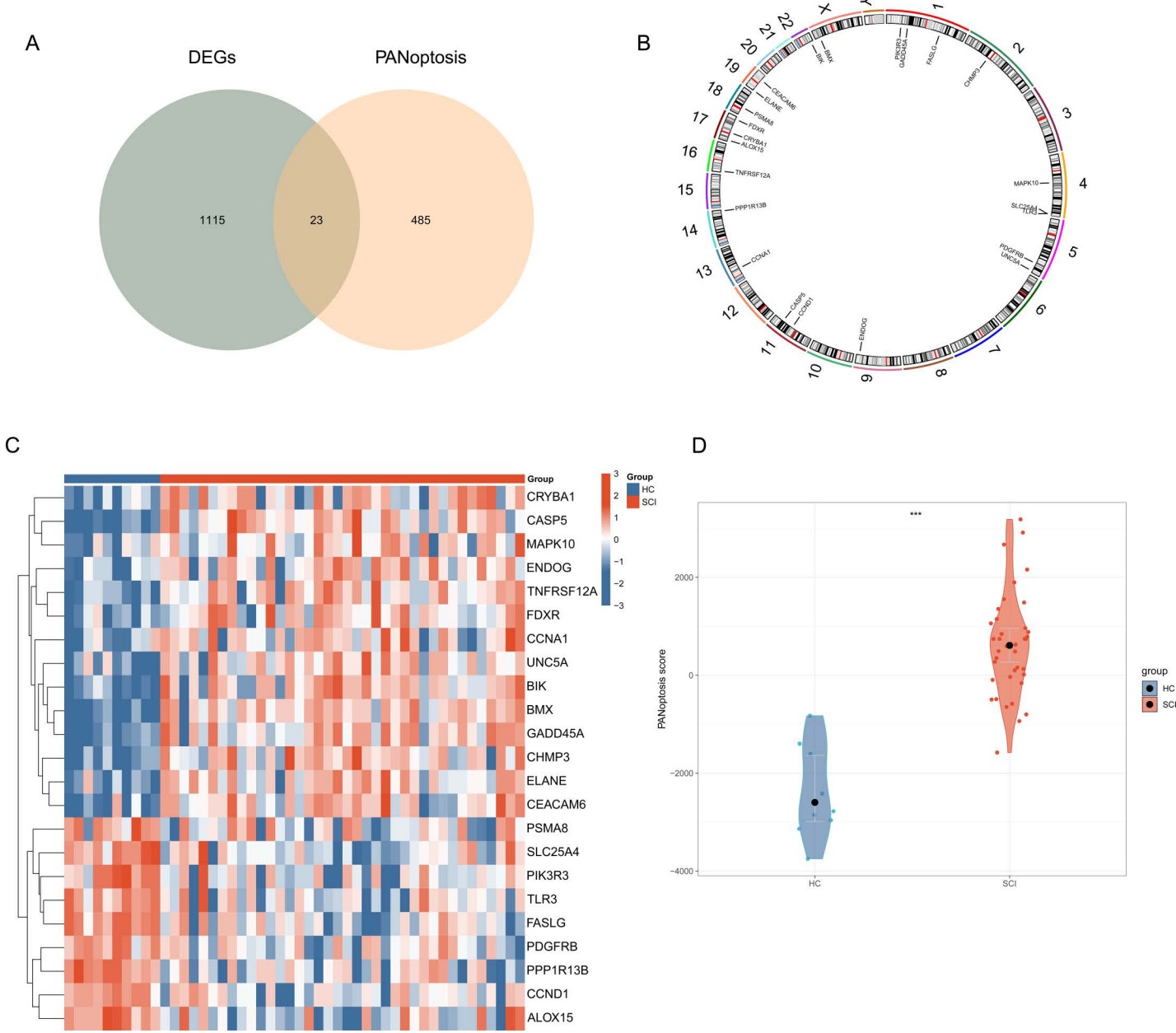

**Fig 3. Identification of PANoptosis-related genes in SCI. (A)** The intersections between DEGs and PANoptosis-related genes. **(B)** Chromosomal positions of the 23 genes. **(C)** Heatmap showing the expression patterns of 23 genes between SCI and HC samples. **(D)** The scores of PANoptosis regulators between SCI samples and HC samples were compared.

for each sample using PCA. The results showed that the PANoptosis score in SCI samples was significantly higher than in HC samples, suggesting PANoptosis contribute to the progression of SCI (Fig 3D).

## GO and KEGG pathway enrichment analysis of common genes

Fig 4 summarizes the top ten GO terms and KEGG pathways. GO annotations encompass three categories: BP, MF, and CC, with no enrichment detected in MF and CC. In the BP process, common genes were mainly enriched in protein kinase activity and protein phosphorylation signaling pathways (Fig 4A). KEGG pathway enrichment analysis included human papillomavirus infection, necroptosis, apoptosis and FoxO signaling pathways (Fig 4B).

## Hub gene identification in SCI and correlation Analysis

To explore protein–protein interactions, we constructed a PPI network for the 32 common genes using the STRING database and visualized the network using Cytoscape (Fig 5A). To identify key hub genes within the network, we

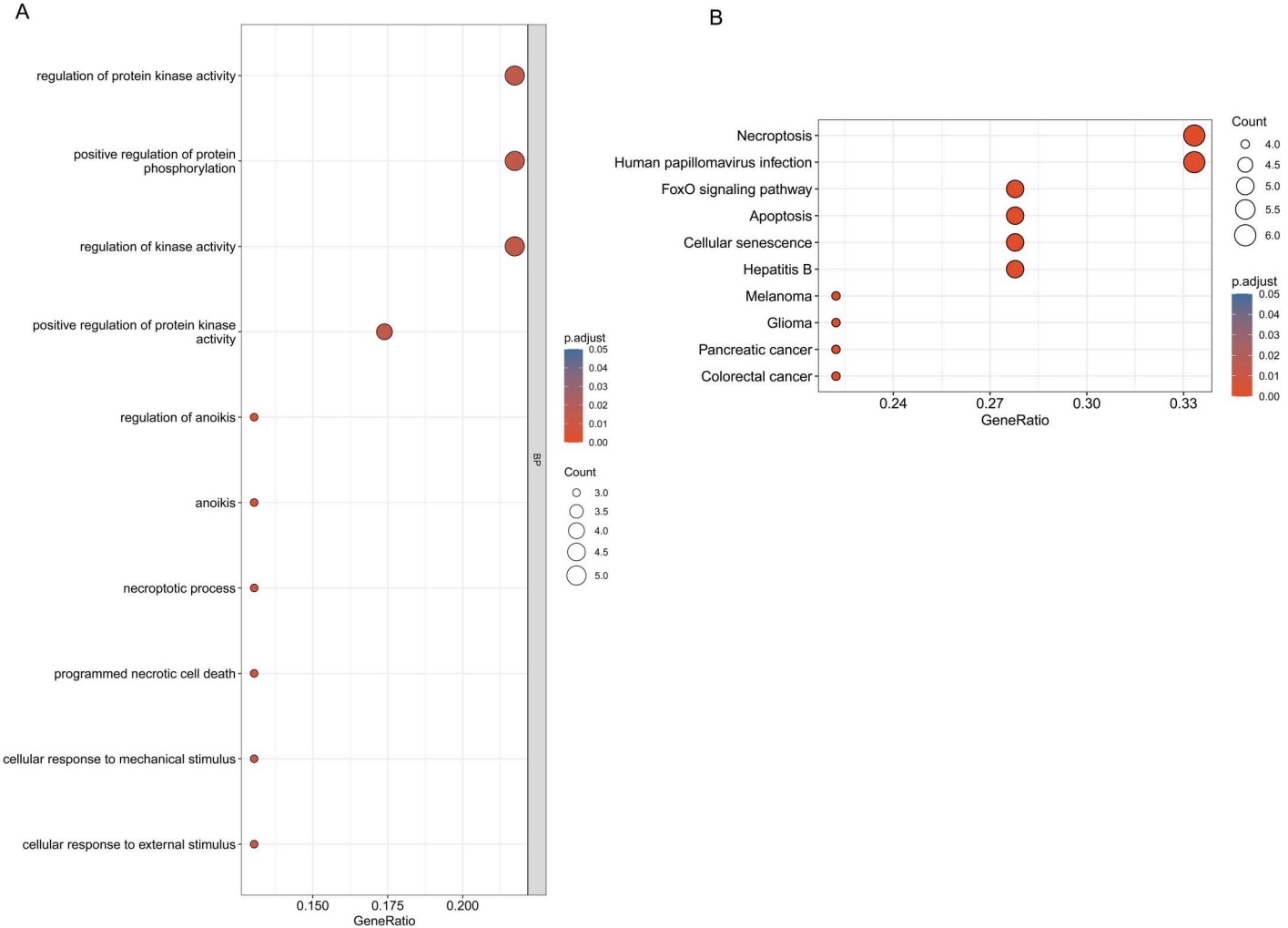

**Fig 4. Functional analysis of the 23 common genes was performed. (A)** The detailed relationship between hub genes and the top 10 pathways annotated by GO functional enrichment analysis. **(B)** KEGG pathway enrichment analysis identified the top 10 pathways enriched in hub genes.

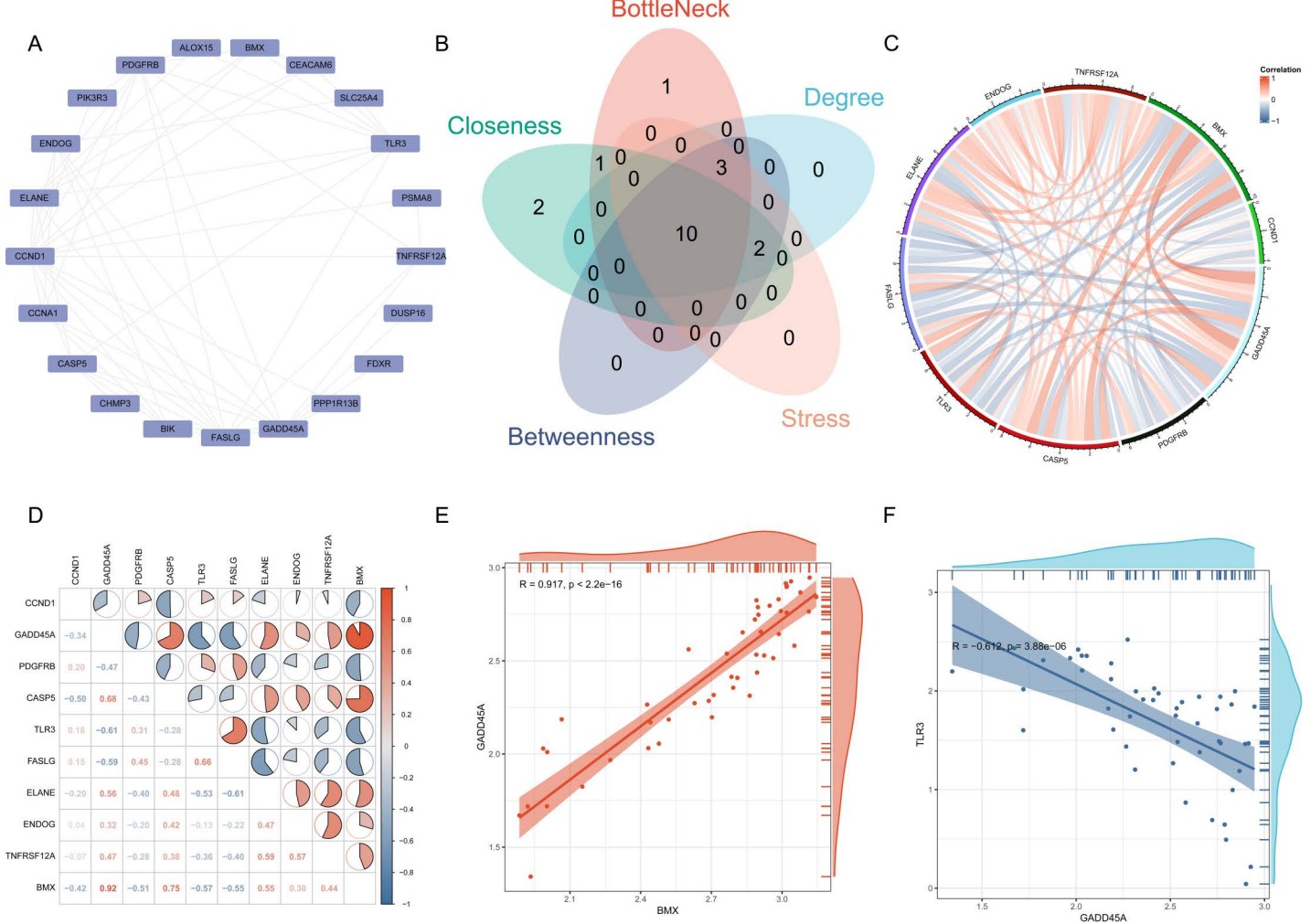

**Fig 5. The PPI network and the interactions among hub genes. (A)** A PPI network was constructed for the 23 common genes. **(B)** Using five algorithms (BottleNeck, Closeness, Degree, Betweenness, Stress) from the cytoHubba plugin, 10 common hub genes were identified. **(C)** Gene relationship circle diagram for the 10 hub genes, with red and blue lines representing positive and negative correlations, respectively. **(D)** The relationship between hub genes, with correlation coefficients indicated by the area of the pie chart. **(E and F)** Scatterplots were used to display the highest correlations among 10 hub genes: *GADD45A* and *BMX* (positive correlation), and *TLR3* and *GADD45A* (negative correlation).

applied five topological analysis algorithms (Degree, Closeness, Betweenness, Stress, and BottleNeck) using the cytoHubba plugin in Cytoscape. The top 15 genes identified by each algorithm were intersected, and a total of 10 genes (*CCND1*, *GADD45A*, PDGFRB, *CASP5*, *TLR3*, *FASLG*, *ELANE*, *ENDOG*, *TNFRSF12A*, and *BMX*) were consistently ranked across multiple methods. The results are presented as a Venn diagram (Fig 5B), illustrating the overlap among the top 15 genes identified by five cytoHubba algorithms, and highlighting the robustness of the 10 hub genes consistently ranked across methods. These genes are likely to serve as central regulators in the molecular mechanisms underlying spinal cord injury. The correlation analysis among the 10 core genes indicated high correlations between these regulators (Figs 5C and 5D), with the highest positive correlation between *GADD45A* and *BMX* (correlation coefficient 0.917, Fig 5E), and a negative correlation between *TLR3* and *GADD45A* (correlation coefficient −0.612, Fig 5F).

## Identification of SCI-associated hub genes using 8 machine learning algorithms

Results yielded by LASSO (Fig 6A), LQV algorithm (Fig 6B), Boruta algorithm (Fig 6C), Bagged Tree algorithm (Fig 6D), Random Forest (Fig 6E), Bayesian (Fig 6F), SVM algorithm (Fig 6G), xgboost algorithm (Fig 6H) identified 2 genes closely associated with SCI pathogenesis (Fig 6I): *BMX* and *CASP5*.

## Immune cell infiltration and correlation analysis of hub genes in SCI

We estimated the relative abundance of 22 immune cells in SCI and normal samples using the CIBERSORT algorithm and the LM22 feature matrix. The results indicated that, compared to the HC group, the SCI group exhibited higher

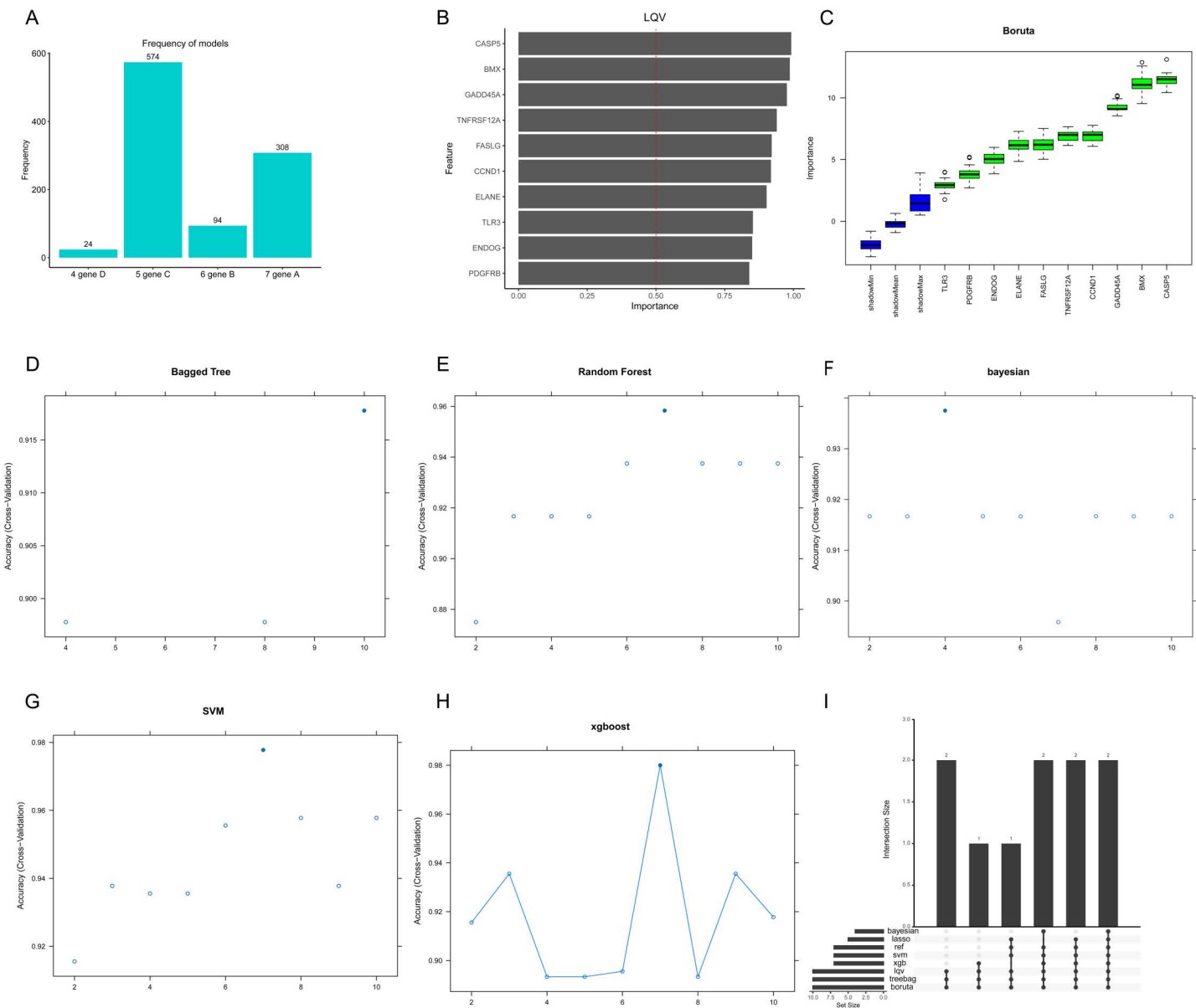

**Fig 6. Machine learning analysis of 10 hub genes related to SCI was performed using eight algorithms applied to the dataset.** The results from different machine learning methods include: Lasso-Logistic regression **(A)**, LQV algorithm **(B)**, Boruta **(C)**, Bagged Tree **(D)**, Random Forest **(E)**, Bayesian **(F)**, SVM (G) and xgboost **(H)**. **(I)** A summary of the genes identified as important for SCI by all eight algorithms.

levels of neutrophils, activated mast cells, monocytes, and activated NK cells, while eosinophils, resting NK cells, resting CD4 memory T cells, and CD8 T cells were less enriched. (Fig 7A). Correlation analysis of immune cells revealed negative correlations between resting CD4 memory T cells and CD4 naive T cells, and positive correlations between CD8 T cells and activated CD4 memory T cells (Fig 7B). Correlation analysis between hub genes and immune cells revealed that *CASP5* was positively correlated only with gamma delta T cells, while *BMX* showed positive correlations with plasma cells, activated dendritic cells, and activated mast cells, but negative correlations with resting CD4 memory T cells, activated CD4 memory T cells, and CD8 T cells (Fig 7C). Compared to *CASP5*, *BMX* exhibited a stronger correlation with immune cell infiltration. Further analysis using a lollipop chart revealed that *BMX* had the highest correlation with Mast cells activated (Fig 7D). A scatter plot further displayed the positive correlation between *BMX* and Mast cells activated (Fig 7E).

### Single-gene ROC analysis of *BMX* in SCI

*BMX* expression is significantly higher in the SCI group compared to the HC group in the gene set (Fig 8A). Based on single-gene ROC analysis, the AUC values for *BMX* were 0.987 (Fig 8B).

### Single-gene GSEA analysis reveals key biological pathways associated with *BMX* in SCI

We performed single-gene GSEA analysis to explore the biological functions and pathways associated with *BMX*. The results showed that *BMX* was positively correlated with metabolic pathways such as vitamin A and carotenoid metabolism, bile acid and salt recycling, epigenetic regulation, estrogen's influence on gene expression, and COVID-19-related lung tissue changes, and negatively correlated with translation processes in gene expression, eukaryotic translation initiation and elongation processes, ribosomes, and reference translation initiation pathways (Figs 9A and 9B).

### Exploration of drug-gene interactions and regulatory mechanisms for *BMX* in SCI

CTD database explored drug-gene interactions and identified existing or potential drugs. Targeting *BMX* may provide a specific treatment strategy. The drug-gene interaction network for these genes is shown in Fig 10A. A total of 38 drugs target *BMX*, which is associated with a large number of predicted drug interactions. Subsequently, the potential regulatory mechanisms controlling the expression of *BMX* were analyzed. RNA-binding proteins (RBPs) play a vital role in regulating RNA metabolism in eukaryotes. RBP related to *BMX* were predicted using the StarBase database, and mRNA-RBP interaction network was constructed and visualized using Cytoscape (Fig 10B). Approximately 58 RBPs were predicted using the starBase online database. Transcription factors (TFs) interacting with *BMX* were obtained from the ChIPBase database, and mRNA-TF interaction networks were constructed and visualized using Cytoscape (Fig 10C), which reveals that the *BMX* gene targets 23 TFs.

### Validation of the marker gene *in vivo*

To evaluate *BMX* gene expression in the rat spinal cord injury (SCI) model, qRT-PCR (Fig 11A), Western blotting (Fig 11B), and immunohistochemistry (Fig 11C) were performed. For detailed information on the Western blotting raw images, please refer to S1 File. The results consistently demonstrated a significant upregulation of BMX expression in the SCI group compared to the Sham group (Fig 11).

## Discussion

SCI is a devastating condition characterized by damage to the spinal cord, leading to profound neurological deficits and a significant reduction in the quality of life for affected individuals [27]. This injury can arise from traumatic events such as vehicular accidents, falls, or sports injuries, as well as from non-traumatic causes including infections, tumors, and degenerative diseases [28]. The pathophysiology of SCI involves both primary and secondary injury mechanisms, where the

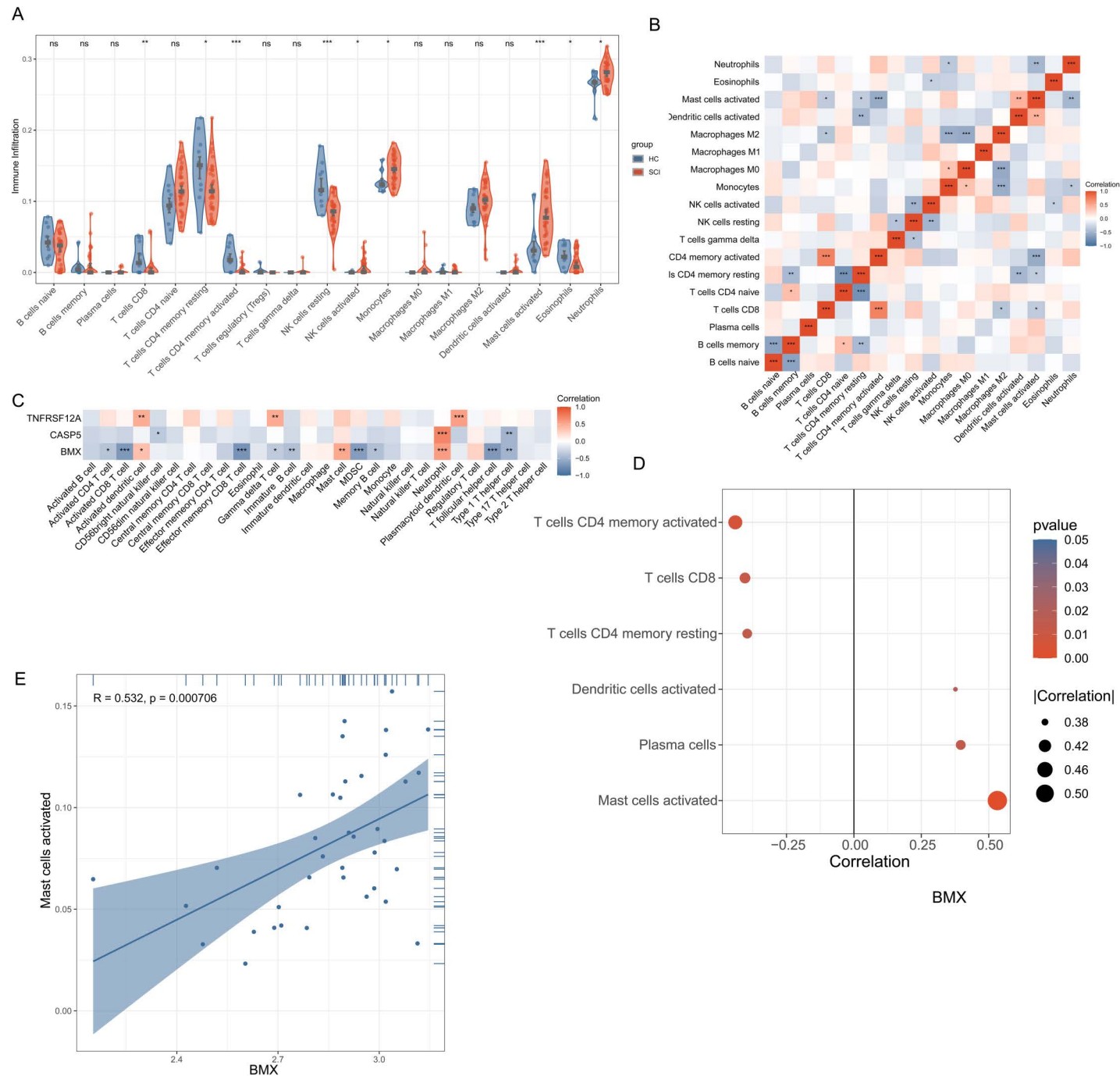

**Fig 7. Immune microenvironment analysis in the SCI and HC groups. (A)** The relative abundance of infiltrated immune cells between SCI and HC. **(B)** Pearson correlation matrix of these immune cells. **(C)** Pearson correlation analysis of the relationship between two hub genes and immune infiltration. **(D)** A lollipop plot was used to visualize the relationship between *BMX* and immune cells. **(E)** Scatterplot was used to display the correlation between *BMX* and Mast cells activated.

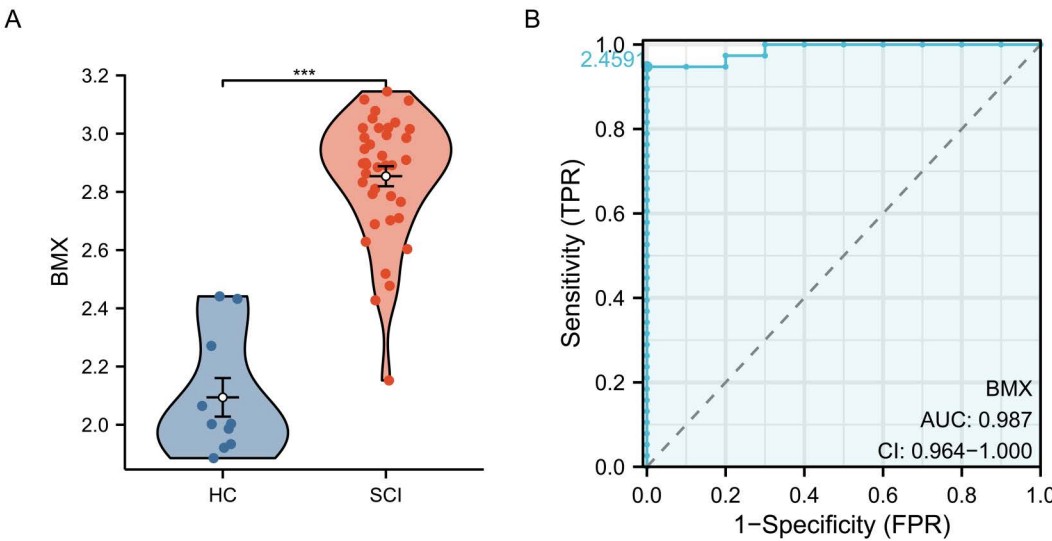

**Fig 8. Expression analysis (A) and ROC curve analysis of *BMX* (B) were performed.**

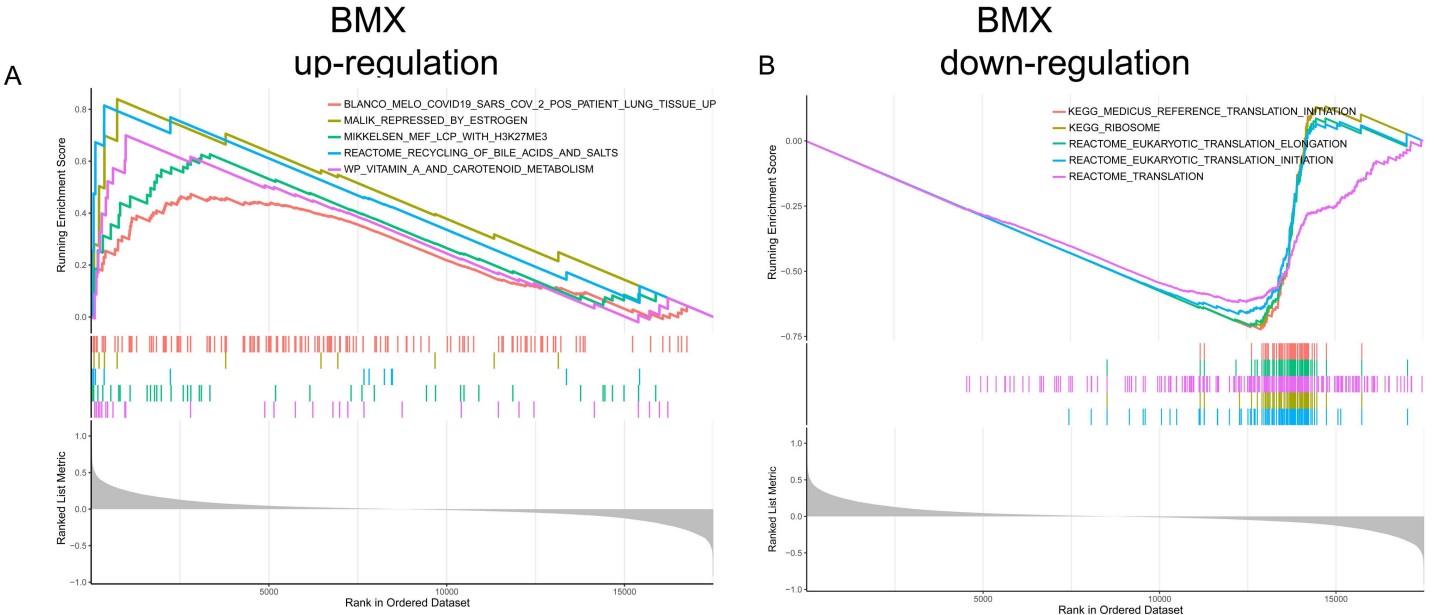

**Fig 9. Pathway and functional analysis of the hub gene. (A)** The top five pathways positively associated with *BMX*. **(B)** The top five pathways negatively associated with *BMX*.

initial trauma disrupts neuronal structures, followed by a cascade of secondary processes that exacerbate damage and hinder recovery [29]. Understanding the multifaceted nature of SCI is crucial for developing effective therapeutic strategies aimed at promoting recovery and improving patient outcomes. In this study, GSEA analysis showed significant enrichment of pathways related to immune regulation, including cytokine-cytokine receptor interactions, immature neutrophil signaling pathways, primary immunodeficiency, neutrophil degranulation, and B lymphocyte progenitor cell signaling pathways.

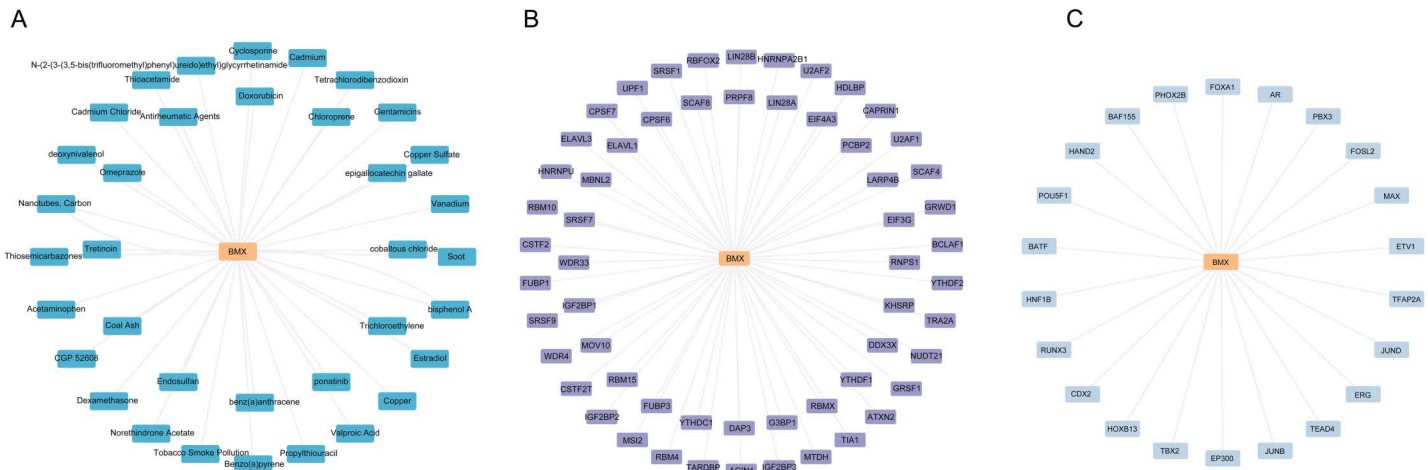

**Fig 10. Drug-gene, RBP-mRNA, and TF-mRNA network construction. (A)** Candidate drug molecules targeting the feature gene. **(B)** Candidate RBP targeting the feature gene. **(C)** Candidate TF targeting the feature gene.

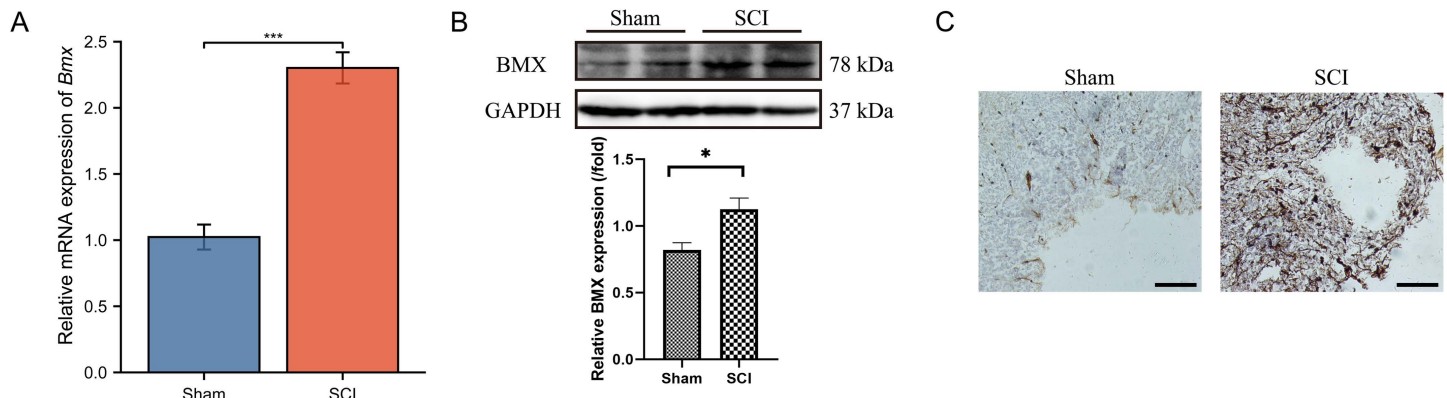

**Fig 11. Experimental verification of BMX expression in SCI. (A)** Relative mRNA expression of *Bmx* in the spinal cord tissue of SCI and sham rat was evaluated by RT-qPCR. **(B-C)** WB analysis was used to detect BMX expression in spinal cord tissues, followed by densitometric quantification. **(D)** Images of IHC staining showing BMX expression in the Sham and SCI groups (scale bar = 50 μm). *$P < 0.05$, **$P < 0.01$, ***$P < 0.001$.

Subsequently, GSVA analysis found that cholesterol homeostasis, reactive oxygen species signaling pathways, IL-6/JAK/STAT3 signaling pathways, glycolysis, and inflammation-related signaling pathways were upregulated in SCI group.

PANoptosis, a novel form of regulated cell death that encompasses apoptosis, pyroptosis, and necroptosis, has been shown to play a critical role in various inflammatory diseases and tissue injuries, suggesting its potential significance in SCI [30]. In this study, we developed a PANoptosis gene set and demonstrated significant activation of PANoptosis in SCI patients. We investigated the role of PANoptosis-related genes in SCI using comprehensive bioinformatics analyses. Our findings elucidate the intricate relationship between PANoptosis and the pathophysiological mechanisms underlying SCI.

The identification of 23 differentially expressed PANoptosis-related genes provides a foundation for further exploration of their functional roles in SCI pathology. Notably, the elevated PANoptosis score in SCI patients highlights the involvement of this cell death pathway in the disease's progression. These common genes were mainly enriched in protein kinase activity, protein phosphorylation signaling pathways, necroptosis, apoptosis and FoxO

signaling pathways. These processes and pathways play a significant role in the development of SCI [31–34]. Moreover, our PPI network analysis revealed identified 10 hub genes. Among these hub genes, *GADD45A* and *BMX* exhibited a positive correlation, while *TLR3* and *GADD45A* showed a negative correlation. Subsequently, eight machine learning algorithms were employed to further refine the selection of genes. *BMX* and *CASP5* were consistently identified across all algorithms.

Immune cells of the innate immune response are essential in the pathophysiology of SCI and inflammation [35]. To better grasp the infiltrated immune cells of SCI and the correlation with *BMX* and *CASP5*, we constructed the immune infiltration analysis and found that *BMX* showed strong associations with immune cell infiltration, highlights the potential for targeting these pathways. *BMX* was positively correlated with Plasma cells, Dendritic cells activated, Mast cells activated, and negatively correlated with T cells CD4 memory resting, T cells CD4 memory activated, T cells CD8.

The spatial distribution and movement of immune cells have a significant impact on the inflammatory response and subsequent healing processes after SCI [3]. Studies have indicated that modulation of PANoptosis can influence inflammatory responses and neuronal survival, suggesting that interventions aimed at regulating this pathway may hold promise for improving functional outcomes in SCI patients [36]. Bone marrow tyrosine kinase on chromosome X (BMX), a non-receptor tyrosine kinase from the Tec family with Src homology (SH) 3, SH2, and carboxy-terminal kinase domains, plays a vital role in processes like cytokine signaling and inflammation [37,38]. Previous research has indicated that BMX contributes to the inflammatory response by modulating Toll-like receptor-induced interleukin-6 (IL-6) production and interacts with Fas to stimulate the release of IL-6 and tumor necrosis factor-α (TNF-α) [39,40]. In this study, *BMX* exhibits a strong positive correlation with Mast cells activated. Mast cells, originating from hematopoietic progenitor cells, mature in vascular tissues and contribute to both innate and adaptive immune responses [41]. Mast cells can secrete a variety of immune and inflammatory mediators, such as histamine, β-tryptase, TNF-α, and interleukin-1β (IL-1β), which play key roles in immune responses [42]. Furthermore, we validated the high expression of BMX in the SCI animal model through qRT-PCR, WB and IHC experiments. These findings suggest that BMX may serve as a novel and potential biomarker for SCI. Therefore, future studies should prioritize validating the association between BMX and mast cells in preclinical SCI models. We performed single-gene GSEA analysis to explore the biological functions and pathways associated with BMX. The results showed that *BMX* was positively correlated with metabolic pathways such as vitamin A and carotenoid metabolism, bile acid and salt recycling, epigenetic regulation, estrogen's influence on gene expression, and COVID-19-related lung tissue changes, and negatively correlated with translation processes in gene expression, eukaryotic translation initiation and elongation processes, ribosomes, and reference translation initiation pathways. These results also suggest that BMX's role in promoting PANoptosis in SCI may regulate the aforementioned biological processes and signaling pathways. However, additional research is needed to confirm the precise role of BMX in SCI and uncover the underlying mechanisms.

Furthermore, the identification of potential therapeutic drugs and transcription factors associated with BMX emphasizes the translational potential of our findings in clinical settings. The integration of bioinformatics data with experimental validation could pave the way for novel targeted therapies aimed at mitigating injury and promoting recovery in SCI. Overall, our study highlights the critical role of PANoptosis-related genes in SCI and sets the stage for future investigations into their potential as biomarkers and therapeutic targets in spinal cord injuries.

In this study, we used a combination of qPCR, Western blot, and immunohistochemistry on SCI animal models to investigate gene and protein expression changes following spinal cord injury. The consistency between qPCR and WB results confirmed that observed gene expression changes were translated into protein alterations. IHC further localized these protein changes within the injured tissue, providing spatial context to the inflammatory response.

The use of animal models allowed us to minimize the heterogeneity of human samples and directly observe the effects of SCI in a controlled environment. These *in vivo* results, combined with bioinformatics analysis of peripheral blood

samples, offer a more comprehensive view of the systemic and localized responses to SCI. Future studies will expand on these findings by using larger cohorts and exploring additional biomarkers to further investigate the complex inflammatory processes following SCI.

The limitations of this study warrant careful consideration. Firstly, while bioinformatics approaches provide valuable insights, they are inherently limited by the quality of the available data and the methods used for analysis. Relying solely on bioinformatics approaches without experimental validation may restrict a comprehensive understanding of the patho-physiological processes underlying SCI. In this study, we utilized the GSE151371 dataset, which contains transcriptomic data from peripheral blood leukocytes of SCI patients and healthy controls. While this dataset provides useful insights into the transcriptional activity, the relevance of these findings to the SCI pathology *in vivo* remains to be fully confirmed. To address this limitation, we have validated the expression of BMX, a key gene identified in our bioinformatics analysis, using experimental methods, including Western blotting, qRT-PCR, and immunohistochemistry, in a rat SCI model. How-ever, further research is needed to confirm the specific role of BMX in SCI and to uncover the underlying mechanisms in a more clinically relevant setting.

Additionally, the sample size derived from the GEO database may not adequately represent the heterogeneity of SCI patients, which could potentially influence the generalizability of the findings. Furthermore, since the experimental valida-tion was conducted using a rat model, there may be species-specific differences that could affect the translation of these findings to human SCI. The use of a single-source dataset may introduce biases that affect the interpretation of results. To address these limitations, future research should incorporate diverse datasets from multiple sources, along with exper-imental validation in both animal models and human patient samples, to further validate and expand upon the findings presented herein.

## Conclusions

This study elucidates significant factors related to the pathogenesis of SCI by employing a multifaceted analytical approach, including GSVA enrichment analysis, differential expression of PANoptosis-related genes, and immune cell infiltration assessment. The identification of BMX and its interactions provides valuable insights into the molecular mech-anisms of SCI. These findings not only enhance our understanding of the disease but also pave the way for the develop-ment of novel diagnostic and therapeutic strategies. However, further validation through experimental studies and larger cohorts is essential to confirm the clinical relevance of the identified biomarkers and pathways.

## Supporting information

**S1 Table. PANoptosis related genes.**
(XLSX)

**S1 File. Western blot raw images.**
(DOCX)

## Author contributions

**Conceptualization:** Xiaoqin Liu, Tianbao Feng.

**Data curation:** Jiating Hu.

**Formal analysis:** Qi Wang.

**Methodology:** Jiating Hu, Mi Xie, Guodong Shi.

**Software:** Jiating Hu, Mi Xie, Guodong Shi.

**Visualization:** Tianbao Feng, Jingyuan Yao.

**Writing – original draft:** Tianbao Feng.

**Writing – review & editing:** Xiaoqin Liu.

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
