## [Decision Letter · Decision Letter 0]

PONE-D-25-09725Identification and experimental validation of BMX as a crucial PANoptosis‑related gene for immune response in Spinal Cord InjuryPLOS ONE

Dear Dr. Liu,

Thank you for submitting your manuscript to PLOS ONE. After careful consideration, we feel that it has merit but does not fully meet PLOS ONE’s publication criteria as it currently stands. Therefore, we invite you to submit a revised version of the manuscript that addresses the points raised during the review process.

Your manuscript was reviewed by two experts and both of them suggested many useful comments which should be addressed during revision.

We look forward to receiving your revised manuscript.

Kind regards,

Partha Mukhopadhyay, Ph.D.

Section Editor

PLOS ONE

Journal Requirements:

4. To comply with PLOS ONE submissions requirements, in your Methods section, please provide additional information regarding the experiments involving animals and ensure you have included details on (1) methods of sacrifice, and (2) efforts to alleviate suffering.

Reviewers' comments:

Reviewer's Responses to Questions

**Comments to the Author**

1. Is the manuscript technically sound, and do the data support the conclusions?

Reviewer #1: Partly

Reviewer #2: Yes

2. Has the statistical analysis been performed appropriately and rigorously? 

Reviewer #1: Yes

Reviewer #2: Yes

3. Have the authors made all data underlying the findings in their manuscript fully available?

Reviewer #1: Yes

Reviewer #2: Yes

4. Is the manuscript presented in an intelligible fashion and written in standard English?

Reviewer #1: Yes

Reviewer #2: Yes

5. Review Comments to the Author

Reviewer #1: The authors used a Database of human patients some hours after spinal cord injury (SCI) to search for genes and/or pathways that could be important for this kind of lesion. I am only slightly familiar with bioinformatics, but I know that the results are as good as the database, and how it is analyzed and interpreted counts as much as the analysis itself.

First, the dataset you are based on is, as you cited, white blood cells of peripheral blood. Although the authors mention justified limitations for this manuscript, I still think some caveats should be addressed. As it is not a dataset with samples from the injured tissue (which, for the nature of the lesion, is very hard to accomplish), an important approach would be selecting cell markers to trace an inflammatory profile of the subjects that could correspond to the changes in the genes you show.

More importantly, I miss more in vivo data. The authors mention the heterogenicity of human samples. Here there was the chance to minimize such effects and to directly study how this lesion would affect individuals in a controlled environment.

When we perform animal experiments, we must use the maximum out of them for ethics sake. Here you brought only one PCR result from the spinal cord of the animals. The very fact that you studied the injured neural tissue in the animals and peripheral WBC in your dataset should be used as a caveat. To ease such discrepancy, you could run some RT-qPCR for inflammatory cell markers and try to relate that to what you find in the human peripheral WBC dataset you used. If the authors still have samples of the injured tissue for histology, immunohistochemistry or in situ hybridization could add better knowledge to your data and a visualization of the cell infiltration.

The in vivo experiments you bring here are not a proper validation of BMX as the authors claim in the title. Better validation of the importance of a single gene to make it worth publication alone could include experimental groups either treated with blockers/antagonists of the gene transcript or knockout animals. The authors mention other genes with possible importance on the way to get to BMX, like the CASP5. An alternative to this issue would be also to check the top 3, top 5 genes that are altered in the bioinformatics analysis.

Nevertheless, I am also curious about why the authors chose to work with female rats. I suppose those were intact animals because you don’t mention any previous surgery. Was there any sort of estrus cycle control made to use the animals at the same time for all animals? Estrogen shows a protective effect on spinal cord injury (PMID: 36736846; 26461840), even therapeutically (PMID: 32680442; 12853305). This should be also raised as a limitation or a possible future direction, comparing males and females.

Minor points:

- Gene names, scientific names, and Latin terms such as in vivo should be in italics.

- Supplementary Table 2 should be incorporated into the manuscript.

Reviewer #2: My comments are uploaded as an attachment and I recommend some minor changes to be done before the final acceptance is made. The manuscript is well-written in terms of global scientific reporting standards but needs some revisions.

6. PLOS authors have the option to publish the peer review history of their article (what does this mean? ). If published, this will include your full peer review and any attached files.

**Do you want your identity to be public for this peer review?** For information about this choice, including consent withdrawal, please see our Privacy Policy .

Reviewer #1: No

Reviewer #2: No

---

## [Author Response · Author response to Decision Letter 1]

26 May 2025

Dear Reviewers,

I would like to express my sincere gratitude for the time and effort you have dedicated to evaluating my manuscript titled "Identification and Experimental Validation of BMX as a Crucial PANoptosis-Related Gene for Immune Response in Spinal Cord Injury." I greatly appreciate the constructive comments and suggestions, which have significantly improved the quality of the paper.

1. ORCiD iD Validation: We will follow the steps outlined in your email to validate our ORCiD iD in the Editorial Manager system as required. We understand the importance of completing this validation and will ensure it is done promptly.

2. Animal Experiment Details: In response to your request, we have updated the SCI rat model establishment section to include additional information regarding the animal experiments. Specifically, the revised text is as follows:

"At the end of the experiments, the animals were euthanized by administering an overdose of isoflurane for approximately 10 minutes, followed by exsanguination."

This update addresses the details of euthanasia and the measures taken to alleviate suffering in accordance with PLOS ONE’s submission guidelines.

In this revised version, I have addressed all the points raised by the reviewers. Below, I provide detailed responses to each comment, including the corresponding changes made to the manuscript. We have also addressed the reviewers’ comments in a separate document titled “Response to Reviewers.” For clarity, the reviewers' comments are included in italicized bold and our specific responses follow in Times New Roman font.

Thank you once again for your valuable feedback and consideration.

Sincerely,

Xiaoqin Liu, PhD

Yan’an Medical College of Yan’an University

---

## [Decision Letter · Decision Letter 1]

PONE-D-25-09725R1Identification and experimental validation of BMX as a crucial PANoptosis‑related gene for immune response in Spinal Cord InjuryPLOS ONE

Dear Dr. Liu,

Thank you for submitting your manuscript to PLOS ONE. After careful consideration, we feel that it has merit but does not fully meet PLOS ONE’s publication criteria as it currently stands. Therefore, we invite you to submit a revised version of the manuscript that addresses the points raised during the review process.

Your revised manuscript was reviewed same experts and one of them suggested a minor revision. Please address those comments and  a quick  editorial decision will be taken after satisfactory revision without sending to experts.

We look forward to receiving your revised manuscript.

Kind regards,

Partha Mukhopadhyay, Ph.D.

Section Editor

PLOS ONE

Journal Requirements:

Reviewers' comments:

Reviewer's Responses to Questions

**Comments to the Author**

1. If the authors have adequately addressed your comments raised in a previous round of review and you feel that this manuscript is now acceptable for publication, you may indicate that here to bypass the “Comments to the Author” section, enter your conflict of interest statement in the “Confidential to Editor” section, and submit your "Accept" recommendation.

Reviewer #1: All comments have been addressed

Reviewer #2: All comments have been addressed

2. Is the manuscript technically sound, and do the data support the conclusions?

Reviewer #1: Yes

Reviewer #2: Yes

3. Has the statistical analysis been performed appropriately and rigorously? 

Reviewer #1: Yes

Reviewer #2: Yes

4. Have the authors made all data underlying the findings in their manuscript fully available?

Reviewer #1: Yes

Reviewer #2: Yes

5. Is the manuscript presented in an intelligible fashion and written in standard English?

Reviewer #1: Yes

Reviewer #2: Yes

6. Review Comments to the Author

Reviewer #1: I appreciate the authors’ thorough and satisfactory responses to my previous concerns. I have only a few minor corrections to suggest regarding Figure 11. In subpanel 11B, a required symbol is missing. Additionally, the figure legend does not describe the significance symbol used. Lastly, I noted a likely error in the description of the IHC, where a reference is made to the “Lamc1 protein,” which appears to be unrelated or included by mistake. Please address these issues in the revised manuscript.

Reviewer #2: The points and questions that were raised by me has been addressed. So do suggest acceptance of the manuscript.

7. PLOS authors have the option to publish the peer review history of their article (what does this mean? ). If published, this will include your full peer review and any attached files.

**Do you want your identity to be public for this peer review?** For information about this choice, including consent withdrawal, please see our Privacy Policy .

Reviewer #1: No

Reviewer #2: No

---

## [Author Response · Author response to Decision Letter 2]

16 Jun 2025

Reviewer #1: I appreciate the authors’ thorough and satisfactory responses to my previous concerns. I have only a few minor corrections to suggest regarding Figure 11. In subpanel 11B, a required symbol is missing. Additionally, the figure legend does not describe the significance symbol used. Lastly, I noted a likely error in the description of the IHC, where a reference is made to the “Lamc1 protein,” which appears to be unrelated or included by mistake. Please address these issues in the revised manuscript.

Response to Reviewer #1:

Response: We sincerely thank the reviewer for their careful evaluation of our manuscript and for acknowledging our thorough responses to previous concerns. We have addressed the minor issues raised as follows:

Missing symbol in subpanel 11B of Figure 11: We have carefully reviewed Figure 11 and added the required symbol to subpanel 11B as suggested. The figure has been updated accordingly.

Figure legend not describing the significance symbol: We have updated the figure legend to include a clear description of the significance symbol used in Figure 11. This change ensures that the figure legend fully corresponds with the content presented in the figure.

Incorrect reference to “Lamc1 protein” in the IHC description: We appreciate the reviewer’s attention to this detail. After reviewing the manuscript, we found that the reference to the “Lamc1 protein” was an error. We have corrected this by changing it to “BMX protein” to accurately reflect the intended reference. The manuscript has been updated accordingly.

---

## [Editor Report · Decision Letter 2]

Identification and experimental validation of BMX as a crucial PANoptosis‑related gene for immune response in Spinal Cord Injury

PONE-D-25-09725R2

Dear Dr. Liu,

We’re pleased to inform you that your manuscript has been judged scientifically suitable for publication and will be formally accepted for publication once it meets all outstanding technical requirements.

Kind regards,

Partha Mukhopadhyay, Ph.D.

Section Editor

PLOS ONE
---

## [Editor Report · Acceptance letter]

PONE-D-25-09725R2

PLOS ONE

Dear Dr. Liu,

I'm pleased to inform you that your manuscript has been deemed suitable for publication in PLOS ONE. Congratulations! Your manuscript is now being handed over to our production team.

Kind regards,

on behalf of

Dr. Partha Mukhopadhyay

Section Editor

PLOS ONE